# Reconstituted virus–nucleus system reveals mechanics of herpesvirus genome uncoating

Alex Evilevitch* and Efthymios Tsimtsirakis

Department of Experimental Medical Science, Lund University, Lund, Sweden

AFM; nucleus; chromatin; mechanics; capsid; Herpes Simplex Virus type 1

**Author for correspondence:**
*Alex Evilevitch,
E-mail: Alex.Evilevitch@med.lu.se

## Abstract

The viral replication cycle is controlled by information transduced through both molecular and mechanical interactions. Viral infection mechanics remains largely unexplored, however, due to the complexity of cellular mechanical responses over the course of infection as well as a limited ability to isolate and probe these responses. Here, we develop an experimental system consisting of herpes simplex virus type 1 (HSV-1) capsids bound to isolated and reconstituted cell nuclei, which allows direct probing of capsid–nucleus mechanics with atomic force microscopy (AFM). Major mechanical transformations occur in the host nucleus when pressurised viral DNA ejects from HSV-1 capsids docked at the nuclear pore complexes (NPCs) on the nuclear membrane. This leads to structural rearrangement of the host chromosome, affecting its compaction. This in turn regulates viral genome replication and transcription dynamics as well as the decision between a lytic or latent course of infection. AFM probing of our reconstituted capsid–nucleus system provides high-resolution topographical imaging of viral capsid docking at the NPCs as well as force volume mapping of the infected nucleus surface, reflecting mechanical transformations associated with chromatin compaction and stiffness of nuclear lamina (to which chromatin is tethered). This experimental system provides a novel platform for investigation of virus–host interaction mechanics during viral genome penetration into the nucleus.

## Introduction

The viral replication cycle is controlled by information transduced through both molecular interactions and mechanical forces (Roos *et al.,* 2007; Pai and Weinberger, 2017; Evilevitch, 2021). Mechanisms regulated by mechanical forces often involve physical interactions that are less specific and therefore less affected by mutations in viral proteins. Knowledge of mechanical transformations in a host cell during viral replication can facilitate discovery of broad-spectrum antiviral targets that are less prone to development of drug resistance (unlike targeting viral proteins, which undergo rapid mutations; Brandariz-Nunez *et al.,* 2020). The field of mechano-virology, however, is largely unexplored. Atomic force microscopic (AFM) force volume mapping (Roos *et al.,* 2007; Schillers *et al.,* 2016) presents an indispensable methodology for investigating viral infection mechanics. Currently, the main effort has been on investigating the mechanics of viral capsids (Ivanovska *et al.,* 2007; Sae-Ueng *et al.,* 2014) and to a more limited extent on cell membrane mechanics during viral entry and exit (Kol *et al.,* 2007). However, for most DNA and a few RNA viruses, the central virus–host interface is the nuclear membrane, which controls the penetration of the viral genome into the host nucleus with its subsequent replication leading to viral spread (Lieberman, 2008; Hennig and O'Hare, 2015; Mettenleiter, 2016). The nucleus is the stiffest organelle in the cell; its mechanics are defined by the perinuclear cytoskeleton, the nuclear lamina and chromatin (Vaziri *et al.,* 2006; Krause *et al.,* 2013; Bigalke and Heldwein, 2016; Stephens *et al.,* 2017). Mechanical transformations in the cell nucleus in response to viral infection remain unknown, since interrogation of nucleus mechanics requires probing the nuclear envelope stiffness without interference from mechanical structures within the cytoplasmic matrix. In this work, we show, for the first time, how AFM can be successfully used to investigate mechanical transformations in the host cell nucleus caused by herpesvirus binding and its genome uncoating, which results in DNA ejection from viral capsid into nucleus. We used human herpes simplex virus type 1 (HSV-1) as a prototype of the nine known human herpesviruses.

As mentioned above, while AFM provides high-resolution imaging combined with mechanical surface mapping at subnano-Newton sensitivity, the challenge is often in the design of an experimental system suitable for mechanical probing, since the AFM cantilever has to be in direct contact with the biological surface in question. In order to investigate the virus–host mechanical interactions that lead to infection, these studies have to be conducted within the cell. Feasibility of such *in situ* mechanical analysis is limited. In this work, we designed an experimental system that allows AFM probing of mechanical transformations in cell-free nuclei associated with HSV-1 capsids under the physiological conditions of the reconstituted cellular environment. Using isolated cell nuclei reconstituted in cytosol and in presence of an ATP-regeneration system (required for capsid attachment and viral genome ejection into a nucleus), we show that we can

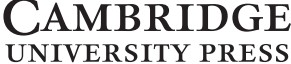

obtain high-resolution AFM topographic imaging individual HSV-1 capsids bound to nuclear pore complexes (NPCs) as well as nano mechanical mapping of the infected nuclear envelope. There are on average over a thousand of NPCs covering a significant area of the nucleus surface (Maul and Deaven, 1977). Viruses binding to NPCs attach to nucleoporins within the NPC structure (Hennig and O'Hare, 2015). We optimised the number of capsids bound to each isolated nucleus to that of a virus/host cell ratio during HSV-1 reactivation from latency in trigeminal ganglia (TG) neuronal cells *in vivo* [100-200 capsids per nucleus (Sawtell, 1997; Sawtell *et al.,* 1998; Thompson and Sawtell, 2000); see details below]. This experimental system can serve as a platform for investigation of nuclear mechanics during viral attachment and genome entry.

In this work, we develop a reconstituted virus–nucleus system for AFM structural analysis of viral capsid docking at the NPCs and probing of nuclear mechanical responses to herpesvirus infection. First, we describe the design of the isolated HSV-1 virus–nucleus system. Next, we demonstrate that this reconstituted nucleus system permits high-resolution AFM imaging of HSV-1 capsids attached to NPCs and offers the capability of direct mechanical mapping of infected nuclei.

## Materials and methods

### Cells and viruses

African green monkey kidney cells [Vero; ATCC CCL-81 from American Type Culture Collection (ATCC), Rockville, MD] and BHK-21 cells (ATCC CCL-10 from ATCC) were cultured at 37°C in 5% $CO_2$ in Dulbecco's modified Eagle's medium (DMEM; Life Technologies, Carlsbad, CA, USA) supplemented with 10% fetal bovine serum (FBS; Gibco), 2-mM L-glutamine (Life Technologies) and antibiotics (100-U $ml^{-1}$ penicillin and 100-µg $ml^{-1}$ streptomycin; Life Technologies). The KOS strain of HSV-1 was used as the wild-type strain. The K26GFP HSV-1 recombinant virus (gift from Dr. Prashant Desai), which carries a GFP tag on the capsid protein VP16, was used in fluorescence studies. All viruses were amplified on Vero cells, and titers were determined on Vero cells by plaque assay. Viral plaque assays were carried out as follows: Viral stocks were serially diluted in DMEM. Aliquots were plated on six-well trays of Vero cells for 1 h at 37°C. The inoculum was then replaced with 40% (v $v^{-1}$) carboxymethylcellulose in DMEM media. HSV-1 plaque assays were incubated for 3–4 days. The monolayers were stained for 1 h with crystal violet stain (Sigma-Aldrich, St. Louis, MO, USA). After removal of the stain, the trays were rinsed with water and dried, and plaques were counted.

### HSV-1 nuclear capsid isolation

Vero cells were grown to confluence and infected with HSV-1 KOS strain at a multiplicity of infection of 5 pfu/cell for 20 h at 37°C. Cells were scraped into solution and centrifuged at 3,500 revolutions per minute (rpm) for 10 min in a JLA-16.250 rotor. The cell pellets were resuspended in phosphate buffered saline (PBS; 1.37-M NaCl, 27-mM KCl, 43-mM $Na_2HPO_4 \cdot 7H_2O$ and 14-mM $KH_2PO_4$), pooled and again centrifuged at 3,500 rpm for 10 min. This washed cell pellet was resuspended in 20-mM Tris buffer (pH 7.5) with protease inhibitor cocktail (cOmplete; Roche, Basel, Switzerland) and incubated on ice for 20 min to swell the cells. The swollen cells were lysed by addition of 1.25% (v $v^{-1}$) Triton X-100 (Alfa Aesar, Haverhill, MA, USA) for 30 min on ice. Samples were centrifuged at 2,000 rpm for 10 min, and the resulting nuclei pellet was resuspended in a small volume of TNE (10-mM Tris, 0.5-M NaCl and 1-mM EDTA) buffer with protease inhibitor cocktail. Nuclei were disrupted by sonication for 30 s (in 10-s intervals, iced between rounds) and large debris were cleared by brief centrifugation (maximum speed for 30 s). $MgCl_2$ and DNase I were added to the supernatant to 20 mM and 100 µg $mL^{-1}$, respectively, and the sample was incubated at room temperature for 20 min. The supernatant was then centrifuged at 11,750 × g for 90 s to pellet large debris, and further cleaned of small debris by underlaying with a 3-mL cushion of 35% sucrose TNE and centrifuging at 23,000 rpm for 1 h. The capsid-rich pellet was resuspended in TNE + protease inhibitor cocktail, then loaded onto a 20–50% (w $w^{-1}$) TNE sucrose gradient and centrifuged at 24,000 rpm in an SW41 rotor for 1 h. The A-, B- and C-capsid bands were extracted by side puncture, diluted at least 3× in TNE buffer and finally centrifuged at 24,000 rpm for 1 h to pellet the capsids. Capsids were gently resuspended in TNE and stored at 4°C. The purification steps for mutant viruses were the same as described for KOS strain.

### Rat liver nuclei isolation and cytosol preparation

Nuclei from rat liver cells were isolated as adapted from previously described protocol (Ojala *et al.,* 2000). The intactness of nuclei was confirmed by light microscopy, EM and FM by staining the nuclei with DAPI and by their ability to exclude fluorescently tagged (Fluorescein isothiocyanate) 70-kDa dextran. The cytosol was separately prepared using BHK-21 cells.

### Reconstituted capsid–nuclei system

An *in vitro* viral HSV-1 DNA translocation system was built in which the HSV-1 genome was released into the nucleoplasm in a homogenate solution mimicking the cytoplasm environment (see details in previously described protocol in Ojala *et al.,* 2000). In a typical system, rat liver cell nuclei were incubated with C-capsids (HSV-1 or GFP-labelled HSV-1), containing: (i) cytosol, (ii) BSA and (iii) ATP-regeneration system (see details in Ojala *et al.,* 2000). These cellular components are required for effective capsid binding to NPCs and opening of the NPC channel for viral DNA translocation (cytosol contains importin-β required for efficient HSV-1 capsid binding to NPCs; Ojala *et al.,* 2000; Anderson *et al.,* 2014). However, presence of these components does not provide an active mechanism for pulling the viral genome across the NPC (Liashkovich *et al.,* 2011; Fay and Pante, 2015; Hennig and O'Hare, 2015), which is instead driven by capsid DNA pressure (Brandariz-Nunez *et al.,* 2019). Prior to AFM measurements, the system was incubated at 37°C for 40 min sufficient for capsid binding to nuclei. For inhibition studies, WGA was pre-incubated with the nuclei prior to addition of C-capsids (Brandariz-Nunez *et al.,* 2019). This experimental setup builds on the previous observation, showing that HSV-1 capsids bind specifically to NPCs of isolated and reconstituted rat liver cell nuclei (Ojala *et al.,* 2000). Furthermore, using pull-down assay combined with qPCR, we showed that HSV-1 capsid–NPC binding in isolated reconstituted nuclei triggers the ejection of viral DNA and its internalisation in the nucleus, driven by capsid DNA pressure of ~18 atm (Brandariz-Nunez *et al.,* 2019). We also separately confirmed that NPCs maintain full transport functionality in the reconstituted nuclei system. This was verified with a fluorescently labelled nuclear localisation signal (NLS; Miyamoto *et al.,* 2002; Brandariz-Nunez *et al.,* 2019). [Purified glutathione S-transferase (GST)–NLS–EGFP recombinant protein, which contains the NLS of the simian virus 40-T antigen fused with GST and EGFP, was used (Miyamoto *et al.,* 2002; Tsuji *et al.,* 2007; Vázquez-Iglesias *et al.,* 2009).]

### Capsid-nucleus sample preparation for AFM

After binding of capsids to nuclei in cytosol reconstituted with ATP-regeneration system, the samples were washed with CBB buffer (20-mM HEPES-KOH with pH of 7.3, 80-mM K-acetate, 2-mM DTT, 1-mM EGTA, 2-mM Mg-acetate, 1-mM PMSF and 1× CLAP cocktail) and stored until usage. The details of substrate and sample preparations can be found elsewhere (Ivanovska *et al.,* 2004; Ivanovska *et al.,* 2007). For fixed nuclei samples, after incubation of capsids with nuclei in cytosol reconstituted with ATP-regeneration system, the samples were washed 2× with CBB buffer (by centrifugation at 3,500 rpm for 10 min at 4°C), then fixed with 2% GA (the pellet was resuspended in 2% GA in PBS) in PBS from 1 to 3 hr. After fixation, the nuclei–capsid samples were washed 2× in CBB buffer, resuspended and placed on the hydrophobically modified glass coverslips and stored until usage.

### Fluorescence microscopy

For fluorescence imaging of the reconstituted capsid–nuclei system, GFP-labelled HSV-1 C-capsids were used. After incubation of capsids with the nuclei as described above, the buffer system containing purified GFP-labelled C-capsids and nuclei were loaded onto coverslips (MatTek, Germany). The nuclei were stained with DAPI for 5 min before imaging. The overlay of the confocal 488 (for GFP emitted signal) and 358 (for DAPI emitted signal) channels shows the localisation of viral capsids onto the nucleus. Images were captured with a Nikon A1R laser-scanning confocal microscope (Nikon Corporation, Japan). For inhibition studies with WGA, the nuclei were pre-incubated with 0.5 mg of WGA ml$^{-1}$ for 20 min on ice before addition of GFP-labelled HSV-1 C-capsids.

### Atomic force microscopy

#### Sample preparation

Nuclei (with or without bound capsids) were dispersed in CBB buffer prior to adsorption by gentle agitation with a pipette. Five microlitre of nuclei suspension was injected into a 50-μl droplet of CBB buffer on an aminoalkylsilane-treated glass substrates and let it to adsorb for 5 min. Afterwards, the sample was gently washed with copious amounts of CBB buffer to remove nonadherent nuclei from the solution. An additional 100 μl of CBB buffer was added after washing.

#### AFM imaging

AFM imaging was performed using a Nanosurf DriveAFM mounted on a Zeiss AxioObserver.Z1 (Zeiss, Germany) or a Nikon TI-S (Nikon, Minato-ku, Japan). The DriveAFM was equipped with an AC40-TS cantilever (Olympus, Shinjuku-ku, Japan) with a nominal spring constant of 0.09 N m$^{-1}$. The actual spring constant of the cantilever was determined using the Sader method followed by contactless determination of the cantilever's deflection sensitivity. Using the optical observation capabilities of the inverted optical microscope, the AFM cantilever was positioned above a nucleus prior to sample approach. AFM imaging was performed in Wave-Mode with a force set point of 600 pN at 2–3-Hz line rate and a typical image size of 512x512 pixels.

#### Force volume mapping

Force mapping was performed with a Nanosurf DriveAFM mounted on a Zeiss AxioObserver.Z1 (Zeiss, Oberkochen, Germany) or a Nikon TI-S. The DriveAFM was equipped with an AC40-TS cantilever with a nominal spring constant of 0.09 N m$^{-1}$.

The actual spring constant of the cantilever was determined using the Sader method followed by contactless determination of the cantilever's deflection sensitivity. The AFM cantilever was positioned above a nucleus prior to surface approach. Force mapping was performed on a 10 × 10 grid of 200nm × 200 nm area. The tip was approached to and retracted from the surface at 300 nm s$^{-1}$ with a force trigger at 250 pN for the approach ramp. Several force maps were captured for each sample. Young's modulus of the nuclei was calculated by applying the Hertz contact mechanics model for a cone indenter to each force curve. Moduli for each force map were plotted in a histogram and the average modulus was assessed using a Gaussian distribution fit. To ensure that nucleus does not roll on the glass substrate surface during AFM indentation, the lateral force was also recorded, showing no lateral drag on the cantilever tip.

## Results and discussion

### Reconstituted virus–nucleus system

Nucleus mechanics in response to viral infection has not been previously investigated. However, attempts were made to investigate nucleus mechanics in a tumour cell environment, where nucle I were probed with AFM on intact cells (Krause *et al.,* 2013). This provided an indirect probing of nucleus stiffness with a colloidal AFM probe (to ensure large surface coverage due to nucleus location uncertainty within a cell) through the cell membrane (Krause *et al.,* 2013). This experimental approach complicates data interpretation, because the nuclear mechanical response is obstructed by the cell membrane and cytoskeleton mechanics (cytoskeleton-mediated tension, where focal adhesion-anchored actin cables pull down the nucleus, thereby partly compressing it; Krause *et al.,* 2013) as well as movement of the nucleus inside the cell. Furthermore, kinetics of capsid trafficking to the nucleus also complicates the analysis (Dunn-Kittenplon *et al.,* 2021). In addition, cytoskeleton-mediated deformation of the nucleus appears as a mechano-transduction pathway through which shear stress may be transduced to a gene-regulating signal (Ingber, 1997; Vaziri *et al.,* 2006). To avoid these perturbations in the experimental approach developed here, we probe nucleus mechanics directly through AFM force volume mapping of a reconstituted capsid–nucleus system. This experimental setup builds on the previous observation that purified HSV-1 capsids bind specifically to NPCs of isolated rat liver cell nuclei reconstituted in cytosol and supplemented with an ATP-regeneration system, which is required for capsid attachment and viral DNA ejection into the nucleus (Ojala *et al.,* 2000); see experimental details in the 'Materials and methods' section. Furthermore, we optimised the number of capsids bound to each isolated nucleus to that of a lytically infected cell, ~150 capsids/nucleus (typical HSV-1 burst size is 100–1,000 virions per infected cell; Zuckerman, 1996) as well as to the infectious virus/host cell ratio during HSV-1 reactivation from latency in TG neuronal cells *in vivo* (occurring at high multiplicity of infection; Sawtell, 1997; Sawtell *et al.,* 1998; Thompson and Sawtell, 2000). A benefit of the reconstituted capsid–nucleus system is that it can be used to isolate the effect of the central step of viral infection (capsid docking at the nucleus and viral genome uncoating) on nucleus mechanics while avoiding interference from other processes occurring within the cell cytoplasm during viral replication.

The HSV-1 capsid assembly process yields capsid assembly intermediates that consist of stably co-purified DNA-filled capsids (C-capsids which are essentially identical to the virion capsid structure; Tandon *et al.,* 2015), and empty capsids (A-capsids and B-capsids) can be isolated (where B-capsids retain cleaved scaffolding proteins; Trus

*et al.,* 1996; Sheaffer *et al.,* 2001; Medina *et al.,* 2012). Purified C-capsids were added to isolated cell nuclei from rat liver cells supplemented with cytosol and an adenosine triphosphate (ATP)-regeneration system and incubated at 37°C. Fig. 1 shows confocal fluorescence microscopic (FM) images of GFP-labelled HSV-1 C-capsids (green, strain K26GFP, HSV-1 strain expressing green fluorescent protein (GFP)-tagged VP26 protein) bound at the surface of DAPI-stained cell nuclei (blue). As a negative control, we used wheat germ agglutinin (WGA), which at high amounts binds the NPCs with high affinity (WGA associates with the glycoproteins within the NPC; Finlay *et al.,* 1987; Ojala *et al.,* 2000) and decreases capsid binding, demonstrating capsid–NPC binding specificity (as opposed to random binding to the nuclear membrane; see Fig. 1). In the next section, we demonstrate the feasibility of AFM analysis for topographical imaging of capsid–nucleus interaction and mechanical mapping of nucleus stiffness during herpes infection.

### Topographical imaging and mechanical mapping of HSV-1-infected nuclei

Direct recording of the nuclear mechanical response to a viral infection has not been previously attempted due to the complexity of interconnected mechanical structures associated with the cell nucleus that are affected by viral replication within the cell (Bigalke and Heldwein, 2016). As mentioned above, the mechanical properties of cells are mostly governed by the joint action of the cytoskeleton (with its three main components: actin filaments, microtubules and the intermediary filaments), the nucleus and the cell membrane (Ingber *et al.,* 2014). Neither has AFM topographical imaging of virus–nucleus interaction been performed on intact nuclei. In previous AFM imaging studies of HSV-1 capsids on the nuclear surface, the nuclear membrane was isolated (the chromatin was removed), fixed with glutaraldehyde (GA) and spread on an AFM substrate surface in order to image structural details of virus attachment on a 2D membrane (Meyring-Wosten *et al.,* 2014). In this work, we show, for the first time, that isolated and reconstituted intact nuclei with HSV-1 capsids adsorbed at NPCs provide a robust experimental system for both AFM structural analysis and interrogation of mechanical transformations associated with the central step of viral infection – viral capsid binding at the nuclear membrane followed by viral genome ejection into the nucleus. Using AFM to resolve high-resolution structures of viral capsids on the nucleus surface requires control of the tip–sample interaction at low force. We show here that this is achieved using an off-resonance AFM imaging mode [WaveMode (Nievergelt *et al.,* 2018), Nanosurf DriveAFM, Liestal, Switzerland; see the 'Materials and methods' section). WaveMode periodically probes the sample surface which allows force control throughout the imaging process (Nievergelt *et al.,* 2018). The photothermally driven sub-resonance actuation of the cantilever allows maintaining controlled, low-force interactions at each point where maximal contact forces are in the pico-Newton range. During each period of the cantilever motion, the tip is completely retracted from the sample surface before moving to

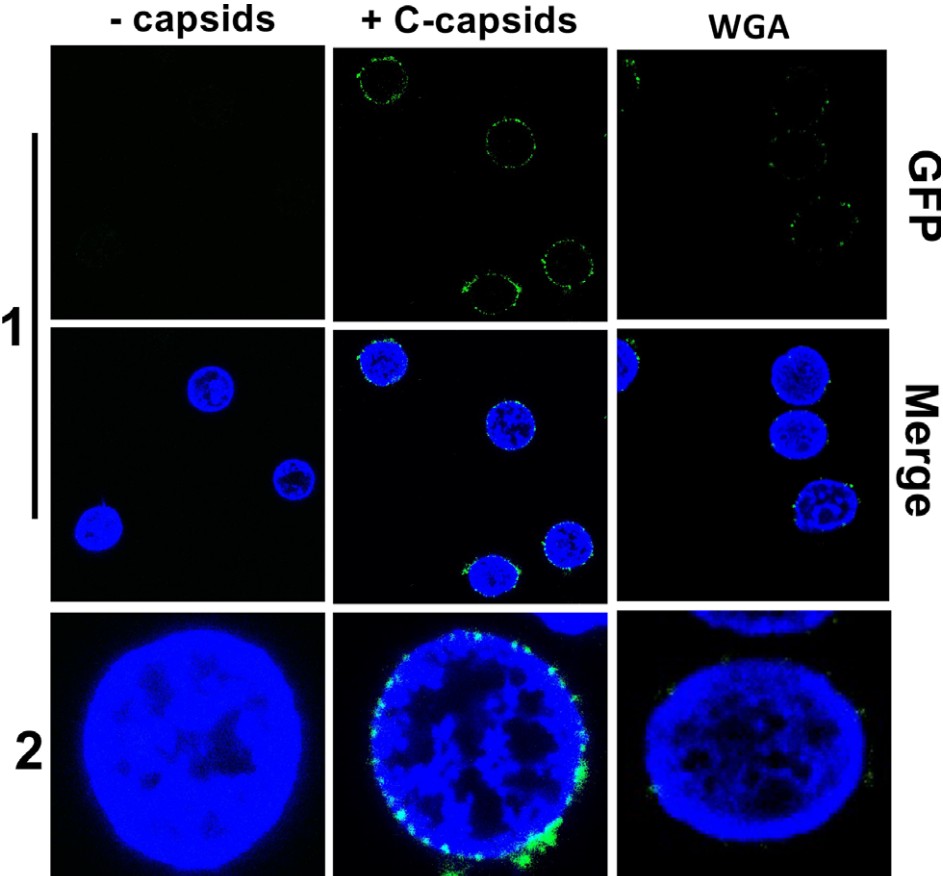

**Fig. 1.** Confocal fluorescence microscopic images show binding of GFP herpes simplex virus type 1 C-capsids (green) to DAPI-stained isolated nuclei (blue), in the presence of cytosol and ATP-regeneration system. The addition of wheat germ agglutinin decreases capsid binding to nuclei for both capsid types, which demonstrates that capsids bind specifically to nuclear pore complexes as opposed to binding anywhere on the nuclear membrane.

the next position, which minimizes lateral forces and sample deformation and thus allows high-resolution topographic imaging. This is in marked contrast to regular tapping mode, in which the interaction force is difficult to control because of the instability of the feedback situation, which can lead to large transient forces with possible sample damage (Schillers *et al.*, 2016).

Another challenge presented by AFM imaging of individual viral capsids on an intact apical nucleus surface is the fact that the nuclear envelope is expected to have a softer mechanical response than the capsid (Sae-Ueng *et al.*, 2014). This suggests that capsids would be pushed into the nucleus surface, resulting in nucleus probing rather than capsid probing (a general AFM imaging requirement is to place a soft object on a hard substrate surface; Schillers *et al.*, 2016). However, as with most biomaterials, nuclei and capsids display viscoelastic behaviour, where apparent stiffness depends on indentation velocity. Indeed, we have found that nucleus stiffness strongly depends on the indentation velocity of the AFM cantilever. Nucleus stiffness is increased with increased tapping velocity, since the nucleoplasm approaches the response of an incompressible medium at a higher indentation velocity (Vaziri *et al.*, 2006; Weber *et al.*, 2019). At the same time, HSV-1 capsids show a predominantly elastic response even at high AFM indentation velocity and therefore do not exhibit strong dependence on AFM tapping velocity (Sae-Ueng *et al.*, 2014). Thus, high AFM probing frequency of the capsid–nucleus surface (we used 8–10-kHz AFM-tip ramp frequency in WaveMode mode) stabilises

the nucleus ('substrate') on which HSV-1 capsids are attached. This further improves imaging resolution. Alternating AFM-tip tapping velocity/ramp rate is a common approach used in AFM microrheology of cells, where modulation of tapping velocity helps to display different nanostructures on the cell surface due to differences in viscoelastic (dynamic Young's modulus value) and elastic (constant Young's modulus value) behaviour of cellular components (Rosenbluth *et al.*, 2006; Rother *et al.*, 2014; Weber *et al.*, 2019).

For topographical imaging of capsids attached to NPCs at the nuclear membrane, purified HSV-1 C-capsids were incubated with isolated nuclei supplemented with cytosol and an ATP-regeneration system. After incubation, the sample was fixated with 2% GA, washed and resuspended in capsid binding buffer (CBB) after which it was deposited on hydrophobically modified AFM slides for imaging (see the 'Materials and methods' section). Fixation of the nuclei in GA cross-links the lamina meshwork (Senda *et al.*, 2005; McKenzie, 2019), which increases nucleus surface stiffness. Since nuclei act as a substrate for capsid imaging, fixation resulting in increased nuclear surface rigidity improves imaging resolution. Fixation, however, is not required for force volume mapping of nucleus surface mechanics. Prior to AFM imaging, nuclei were located with help of an inverted optical microscope to which the AFM was mounted (*Fig. 2a,b,g,h*, showing representative images of two different nuclei located next to the AFM cantilever). First, the AFM was used to image the surface topography of the isolated nucleus without viral capsids in CBB buffer

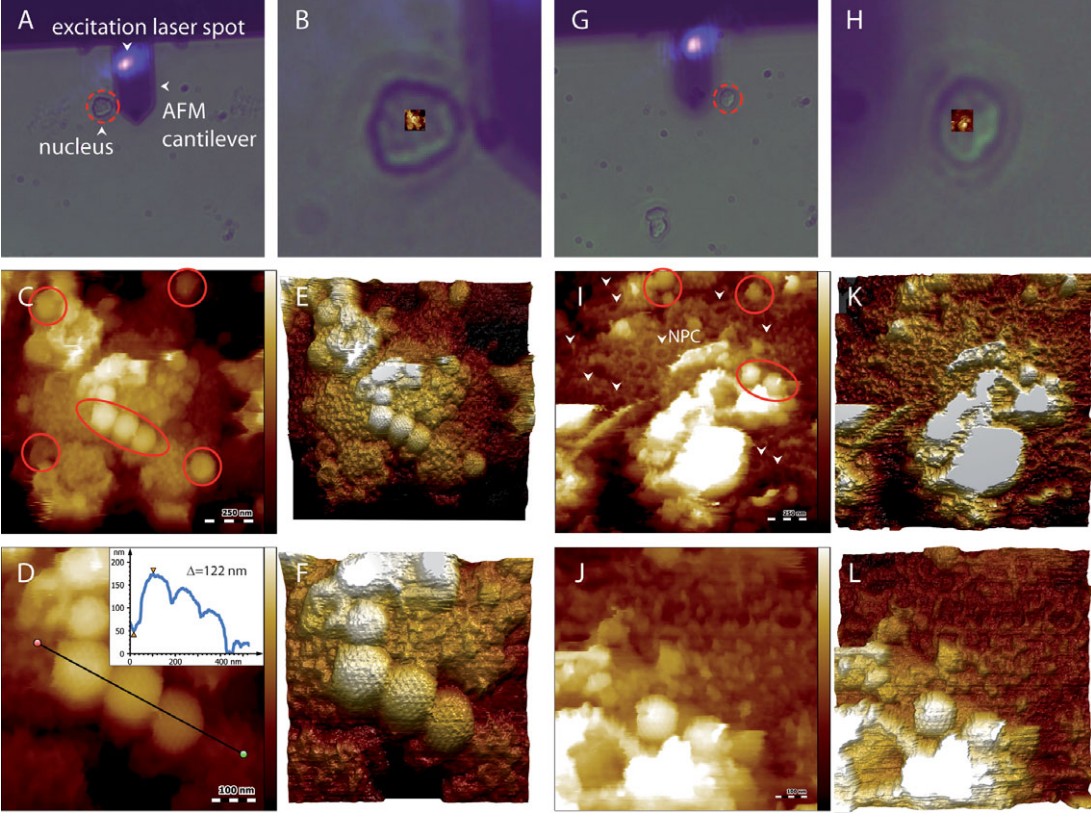

**Fig. 2.** (*a,b*) and (*g,h*) show an overview and a zoomed image of two representative nuclei investigated by atomic force microscopic (AFM) imaging (both nuclei and AFM cantilever with focused laser spot, blue, are visible). In (*b,h*), the imaging area is indicated by the overlayed image. (*c*) shows the topography recorded on the nucleus shown in (*a,b*). Capsids bound to the nuclear surface are encircled in red. (*d*) shows the features marked with an ellipse in (*c*) at higher resolution. The inset in (*d*) shows the surface profile of herpes simplex virus type 1 capsids on the nucleus surface along the black solid line. The height of one capsid is measured to be ~122 nm. (*e,f*) are 3D representations of (*c,d*). (*i*) shows the surface topography of the second nucleus reported in (*g,h*). Virus capsids are encircled in red. Arrows in (*i*) indicate nuclear pore complexes in the nuclear membrane. (*j*) shows a close-up view of the capsids marked with the ellipse in (*i*). (*k,l*) are 3D representations of (*i,j*).
*Note*: The colour scales correspond to 500 (*c*), 300 (*d*) and 250 nm (*i,j*).

(see the 'Materials and methods' section, data not shown). Then, capsid-nucleus assemblies were imaged by AFM (Fig. 2b,h shows an AFM-image of a small area of the nucleus surface overlayed on top of the optical image of a nucleus.) Then, nuclei were incubated with HSV-1 capsids at 37°C for 40 min and washed in CBB buffer. Liquid sample was deposited on the AFM substrate for imaging. Fig. 2c–f (nucleus 1) and i–l (nucleus 2) shows representative 3D topography AFM images of the nuclear membrane surface with well-resolved HSV-1 capsids docked at the NPC baskets. The nucleus surface of rat liver cells is densely covered with protruding NPC baskets, providing binding sites for HSV-1 capsids (Fig. 2i). Capsids on the nucleus surface are displaying faceted icosahedral features as well as resolved individual capsomer subunits characteristic of HSV-1 capsid structure. The capsid height measured on top of the nuclear membrane surface is ~122 nm (Fig. 2d), which is in agreement with a cryo-electron microscopy (EM)-obtained value of 125 nm for HSV-1 capsid diameter (Newcomb et al., 2003). Our previous data with AFM topography of purified HSV-1 capsids alone in buffer solution on a hydrophobic glass surface showed analogous capsid features (Sae-Ueng et al., 2014). This demonstrates that the reconstituted capsid–nucleus system is suitable for AFM structural analysis and, therefore, for mechanical characterisation of infected nuclei.

As mentioned above, a nucleus' mechanical response is dominated by three components – chromatin (where the DNA condensation status reflects on the nucleus stiffness), nuclear lamina (primarily lamin A/C forming thick layers under the nuclear membrane and providing nucleus rigidity) and indirectly by cytoskeleton attachment to the nucleus exterior (which takes part in mechanical signal transduction; Krause et al., 2013; Hobson et al., 2020). Since we are probing mechanics of isolated nuclei, interference from cytoskeleton mechanics is avoided. Chromatin resists strain in nuclear volume, while lamin A/C resists strain in surface area. As mentioned above, in previous nucleus mechanics measurements performed in intact cells (Krause et al., 2013; Hobson et al., 2020), it was suggested that strain in nucleus volume occurs at small AFM indentations (<3 μm, while the average nucleus diameter is ~5–6 μm), where stiffness response is controlled by chromatin; the strain in nucleus area occurs at large indentations (over 50% deformation of nucleus height), where stiffness is controlled by A/C-type lamins. Therefore, in this method development study, we use small indentations relative to nucleus dimension aimed at probing nuclear chromatin's mechanical response with HSV-1 capsids attached at the nuclear membrane. We performed our force volume mapping experiment on an unfixed nucleus surface in CBB buffer with HSV-1 C-capsids attached (after incubation in cytosol supplemented with an ATP-regeneration system at 37°C prompting intranuclear DNA ejection; Brandariz-Nunez et al., 2019). The nucleus indentation force was adjusted to achieve approximately 20–30 nm of nucleus indentation (corresponding to small indentations in comparison to its size; Stephens et al., 2017). Force volume mapping uses lower AFM-tip indentation velocity of 300 nm s$^{-1}$ (corresponding to 0.15 Hz probing frequency), which we determined to be the equilibrium indentation rate below which the modulus of nuclei does not change, in comparison to 8–10 kHz in WaveMode. WaveMode is mainly used for imaging, since indentation velocity varies through the sinusoidal oscillation of the cantilever, while, for force volume mapping, the indentation velocity is kept constant, providing accurate acquisition of the mechanical response. A 10 × 10 force volume map (corresponding to 100 force–distance curves) was acquired over a 200 nm × 200 nm area at the centre of each nucleus where the surface is typically the flattest. Fig. 3a shows a representative force map recorded on an

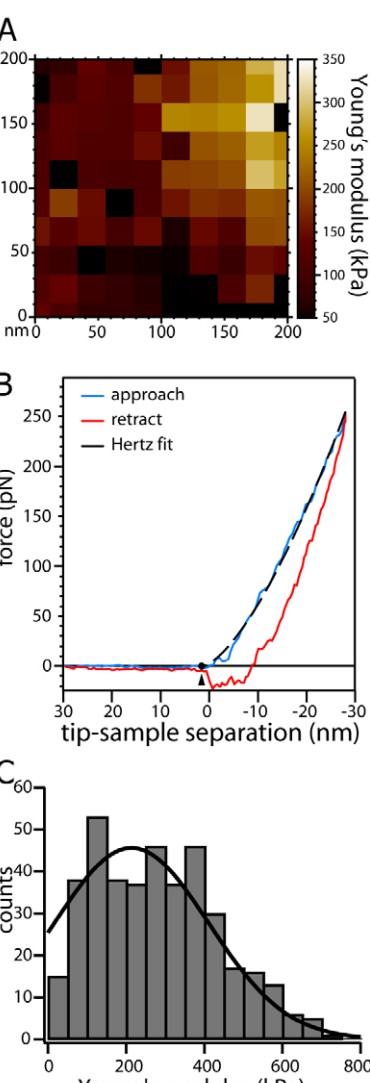

**Fig. 3.** (a) A representative 10 × 10 force volume map for an unfixed nucleus with attached C-capsids (corresponding to 100 force–distance curves) acquired over a 200 nm × 200 nm area at the centre of each nucleus. (b) shows representative force–distance (Fz) curve, comprising each force volume map, for nucleus indentation with herpes simplex virus type 1 C-capsids attached (solid blue line). The red line shows Fz curve for atomic force microscopy (AFM)-tip retraction. The black dashed line shows a fit using the Hertz mechanics model for Young's modulus determination. (c) Young's moduli for 100 force–distance curves in each force volume map were repeatedly collected for several nuclei as well as for several areas on the same nucleus. All moduli values were collectively plotted in a histogram for nuclei with attached C-capsids. The Gaussian fit to the data in the histograms yields the average value and the standard error.

unfixed nucleus with docked C-capsids. Several force maps were captured for each sample. The Young's modulus of the nuclei (elastic modulus describing the amount of strain and reflecting the stiffness) was calculated by applying the Hertz contact mechanics model for a cone indenter to each force curve (Lin et al., 2007; Lin et al., 2007; see the 'Materials and methods' section). Fig. 3b (blue curve) shows a representative force–distance (Fz) cantilever approach curve, for indentation of a nucleus with HSV-1 C-capsids attached, along with the Hertz model fit of Young's modulus (dashed black line). Young's moduli for 100 force–distance curves in each force map were repeatedly collected for several nuclei as well as on several areas of the same nucleus. All Young's moduli values were collectively plotted in a histogram for nucleus with attached

C-capsids (Fig. 3*c*). The Gaussian fit to the data in the histograms yields the average value and the standard error. The measured average Young's modulus value was $214 \pm 10$ kPa. This analysis demonstrates that our AFM approach combined with the cell-free reconstituted nuclei system can be successfully used in future mechanical studies of nucleus and nuclear chromatin mechanics in response to viral capsid docking at NPCs with intranuclear viral DNA ejection.

## Concluding remarks

We show in this work that HSV-1 capsids bound to isolated and reconstituted cell nuclei are an optimal experimental system for AFM structural topographic imaging and force mapping of capsid–nucleus interaction mechanics. AFM force mapping of a herpes capsid–nucleus system can provide information on mechanically transduced mechanisms regulating viral genome replication dynamics (work in progress). The experimental virus–host system developed in this work, combined with AFM force mapping and off-resonance imaging, provides a novel platform for mechano-analysis of viral genome replication dynamics in the course of infection.

**Acknowledgements.** We would like to thank Christian Bippes and David Morgan from NanoSurf for technical assistance with AFM imaging. We would also like to thank Jose Ramon Villanueva for assistance with sample preparation. This work was supported by the Swedish Research Council grants (VR) 349-2014-3962 and 2019-05192 (all to A.E.), and by the Mats Paulsson Foundation to A.E.

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
