## [Reviewer Report]

*Comments to Author*: The manuscript "Reconstituted virus-nucleus system reveals mechanics of Herpesvirus genome uncoating" by Evilevitch and Tsimtsirakis is an interesting AFM methodology paper on the imaging of viral particles docked to the nuclear membrane. Half of the novelty of the Evilevitch and Tsimtsirakis manuscript is using intact (although aldehyde fixated) nuclei, in contrast to earlier studies typically using two-dimensional pieces of the membrane. Half of the novelty is the possibility of force-mapping an infected free-floating nucleus using AFM (although the authors do not compare with a non-infected nucleus). Taken together, I think the paper reports a nontrivial innovation and has good potential leading to future innovation, and thus fits publication in QRB-D with minor adjustments.

List of comments:

1. The Introduction is somewhat short, and Results and Discussion section has an excessive number of references to new material. This is probably because many of the paragraphs in Results and Discussion are of theoretical nature which could be moved to Introduction/Background, preferably arranged into subheadings. Examples: "The nucleus is the stiffest organelle in the cell...", "There are on average over a thousand of NPCs covering a significant area of the nucleus surface..." Also, some of the sentences could probably be merged into Materials and Methods, for example: "Prior to AFM analysis, nuclei were first reconstituted in a cell cytosol solution with added ATP-regeneration system and then incubated for 40 min at 37°C with purified HSV-1 DNA-filled capsids."

2. It is mentioned that a part of the aim of this manuscript is to study the injection of DNA into the nucleus and the associated changes in mechanical properties. It is also mentioned that DNA-free A-capsids are used as a negative control for the normal C-capsids. Shouldn’t force mapping data be presented for both A and C? As of now, Fig 4 is a mapping of DNA-free A-capsids. If space really is an issue, data for the C-capsids should have priority over the control experiment.

3. It seems from Figure 4 that there could be interesting features behind the blurry pixels. Could the image possibly show a capsid, for example? Is it possible to refine such a 10x10 image to 100x100 or more? Even better, an edge finding algorithm could do multiple iterations, adding AFM indentations (pixels) where the contrast is high and ignoring parts where the contrast is low.

4. It seems the force-mapping method is called Fast Force Map mode (FFM), but then on page 6, "Note that force mapping uses lower indentation velocity, ~0.15 Hz (300 nm/s), versus the 500 Hz we used in FFM mode" and "FFM is mainly used for imaging" could lead to confusion. Is the method used to obtain Figure 4 called FFM, or a variant of FFM, or not FFM?

---

## [Reviewer Report]

*Comments to Author*: Reviewer #1: The manuscript "Reconstituted virus-nucleus system reveals mechanics of Herpesvirus genome uncoating" by Evilevitch and Tsimtsirakis is an interesting AFM methodology paper on the imaging of viral particles docked to the nuclear membrane. Half of the novelty of the Evilevitch and Tsimtsirakis manuscript is using intact (although aldehyde fixated) nuclei, in contrast to earlier studies typically using two-dimensional pieces of the membrane. Half of the novelty is the possibility of force-mapping an infected free-floating nucleus using AFM (although the authors do not compare with a non-infected nucleus). Taken together, I think the paper reports a nontrivial innovation and has good potential leading to future innovation, and thus fits publication in QRB-D with minor adjustments.

List of comments:

1. The Introduction is somewhat short, and Results and Discussion section has an excessive number of references to new material. This is probably because many of the paragraphs in Results and Discussion are of theoretical nature which could be moved to Introduction/Background, preferably arranged into subheadings. Examples: "The nucleus is the stiffest organelle in the cell...", "There are on average over a thousand of NPCs covering a significant area of the nucleus surface..." Also, some of the sentences could probably be merged into Materials and Methods, for example: "Prior to AFM analysis, nuclei were first reconstituted in a cell cytosol solution with added ATP-regeneration system and then incubated for 40 min at 37°C with purified HSV-1 DNA-filled capsids."

2. It is mentioned that a part of the aim of this manuscript is to study the injection of DNA into the nucleus and the associated changes in mechanical properties. It is also mentioned that DNA-free A-capsids are used as a negative control for the normal C-capsids. Shouldn’t force mapping data be presented for both A and C? As of now, Fig 4 is a mapping of DNA-free A-capsids. If space really is an issue, data for the C-capsids should have priority over the control experiment.

3. It seems from Figure 4 that there could be interesting features behind the blurry pixels. Could the image possibly show a capsid, for example? Is it possible to refine such a 10x10 image to 100x100 or more? Even better, an edge finding algorithm could do multiple iterations, adding AFM indentations (pixels) where the contrast is high and ignoring parts where the contrast is low.

4. It seems the force-mapping method is called Fast Force Map mode (FFM), but then on page 6, "Note that force mapping uses lower indentation velocity, ~0.15 Hz (300 nm/s), versus the 500 Hz we used in FFM mode" and "FFM is mainly used for imaging" could lead to confusion. Is the method used to obtain Figure 4 called FFM, or a variant of FFM, or not FFM?